# Symmetry-protected hierarchy of anomalous multipole topological band gaps in nonsymmorphic metacrystals

Xiujuan Zhang[1,5], Zhi-Kang Lin[2,5], Hai-Xiao Wang[2,3,5], Zhan Xiong[2], Yuan Tian[1], Ming-Hui Lu �micID [1,4]*, Yan-Feng Chen[1,4]* & Jian-Hua Jiang[2]*

Symmetry and topology are two fundamental aspects of many quantum states of matter. Recently new topological materials, higher-order topological insulators, were discovered, featuring bulk–edge–corner correspondence that goes beyond the conventional topological paradigms. Here we discover experimentally that the nonsymmorphic $p4g$ acoustic meta-crystals host a symmetry-protected hierarchy of topological multipoles: the lowest band gap has a quantized Wannier dipole and can mimic the quantum spin Hall effect, whereas the second band gap exhibits quadrupole topology with anomalous Wannier bands. Such a topological hierarchy allows us to observe experimentally distinct, multiplexed topological phenomena and to reveal a topological transition triggered by the geometry transition from the $p4g$ group to the $C_{4v}$ group, which demonstrates elegantly the fundamental interplay between symmetry and topology. Our study demonstrates that classical systems with controllable geometry can serve as powerful simulators for the discovery of novel topological states of matter and their phase transitions.

[1] National Laboratory of Solid State Microstructures and Department of Materials Science and Engineering, Nanjing University, Nanjing 210093, China. [2] School of Physical Science and Technology and Collaborative Innovation Center of Suzhou Nano Science and Technology, Soochow University, 1 Shizi Street, Suzhou 215006, China. [3] School of Physical Science and Technology, Guangxi Normal University, Guilin 541004, China. [4] Collaborative Innovation Center of Advanced Microstructures, Nanjing University, Nanjing 210093, China. [5] These authors contributed equally: Xiujuan Zhang, Zhi-Kang Lin, Hai-Xiao Wang. *email: luminghui@nju.edu.cn; yfchen@nju.edu.cn; jianhuajiang@suda.edu.cn

Symmetry and topology are two fundamental paradigms in the classification of matters. The fundamental interplay and relation between symmetry and topology have been intriguing for physicists for decades, since the discovery of the quantum Hall effects. Historically, the first model demonstrating the beautiful relation between symmetry and topology is the Su–Schrieffer–Heeger model[1], which shows that quantized dipole polarization of an insulator with chiral symmetry has nontrivial consequences on the edges[2] and serves as a fundamental example of charge fractionalization due to symmetry and topological principles[3].

In quantum mechanics, the bulk dipole polarization of a crystalline insulator is quantified through the Berry phase of the filled Bloch bands[4,5]. In the Bloch–Wannier representation, such a dipole polarization is calibrated by the displacement of the Wannier center with respect to the center of the unit-cell. When generalized to two-dimensional (2D) and three-dimensional (3D) insulators, such quantum description of the dipole polarization can be connected, respectively, to the Hall conductance and magneto-electric polarizability, which characterize the topological responses of the 2D quantum Hall insulators and the 3D time-reversal invariant topological insulators, respectively[6–8]. Recently, quadrupole and octupole topological insulators were proposed[8,9], which extend band topology from dipole polarization to quadrupole and octupole polarizations in the Bloch–Wannier representation. For instance, a 2D quadrupole topological insulator (QTI)[8–12] supports gapped edge states with quantized edge polarizations and in-gap corner states at the edge-terminating corners, demonstrating the higher-order band topology[8–23]. A hallmark of the quadrupole topology is that, counterintuitively, the corner charge is not an addictive function of the edge polarizations but determined solely by the bulk quadrupole topology. As a consequence, each corner hosts only one locacalized mode with a fractional charge of $\frac{1}{2}$, despite that the polarizations of the edges terminated at the corner add up to a trivial corner charge of $0$[8,9].

The emergence of quadrupole topology in a crystalline insulator requires a few fundamental conditions. In the literature[8–12,23], these conditions are as follows: (i) a pair of mirror symmetries that quantize the Wannier dipole and Wannier quadrupole, and make the former vanishing, and (ii) at least two occupied bands for the realization of cancelling dipole moments. The QTI proposed in ref. [8] is based on a square-lattice tight-binding model with $\pi$-flux per plaquette, which needs both positive and negative nearest-neighbor couplings. Such requirements impose challenges for experimental realizations. Up till now, only a few physical systems have realized the QTI phase[10–12]. The 3D octupole topological insulator is even more challenging to be realized due to its complex tight-binding configurations.

In this study, using 3D-printed acoustic metamaterials with controllable geometry, we demonstrate a pathway toward topological multipoles. We use a symmetry-based approach to achieve such a goal, where the $p4g$ crystalline symmetry plays an essential role. We find that 2D sonic crystals (SCs) with the $p4g$ crystalline symmetry can realize a symmetry-protected hierarchy of topological Wannier multipoles without fine-tuning (in fact, even without the guidance from any tight-binding model). In our SCs, the lowest acoustic band gap has a quantized Wannier dipole (denoted as the "dipole topological band gap" in Fig. 1a), whereas the second band gap has an anomalous Wannier quadrupole (denoted as the "quadrupole topological band gap" in Fig. 1a), which cannot be described by any existing theoretical model (but can be verified using Wannier bands and various other characteristics; see Supplementary Notes 1–4). In the literature, the topological Wannier dipole and Wannier quadrupole are

quantized by the mirror symmetries. Counterintuitively, here, the topological Wannier multipoles are quantized without mirror symmetry but by the nonsymmorphic glide symmetries. Moreover, the anomalous quadrupole topology can be annihilated when the symmetry of the SC is tuned from the nonsymmorphic $p4g$ group to the $C_{4v}$ point group by controlling the geometry of the acoustic metacrystal. The quadrupole band topology vanishes exactly at the symmetry transition point, which thus illustrates directly the fundamental interplay between symmetry and topology. We observe for the first time the symmetry-protected hierarchy of topological multipoles in acoustic metacrystals by changing the acoustic frequency, which allows multiplexing topological phenomena as benefited by the fact that there is no fermionic band-filling in acoustic systems.

## Results

**Acoustic metacrystals.** The square-lattice SC (lattice constant $a = 2$ cm) has four arch-shaped scatterers in each unit-cell, which are made of photosensitive resin (bulk modulus 2765 MPa, mass density 1.3 g/cm$^3$, serving as "hard walls" for acoustic waves). The SC is fabricated using the commercial 3D-printing technology (see Methods). The geometries of the four scatterers are identical and are characterized by the arch height $h$, the arm length $l$, and the arm width $w$, which can be tuned to tailor the acoustic bands and their topology (Fig. 1a). Those scatterers are arranged in such a way that the SC has two orthogonal glide symmetries, $G_x = \{m_x | \tau_y\}$ and $G_y = \{m_y | \tau_x\}$ where $m_x := x \to \frac{a}{2} - x$, $m_y := y \to \frac{a}{2} - y$, $\tau_y := y \to y + \frac{a}{2}$ and $\tau_x := x \to x + \frac{a}{2}$ with $a$ being the lattice constant. With the inversion (i.e., parity) and the $C_4$ rotation symmetries, the system has a nonsymmorphic space group of $p4g$. Plastic cladding boards above and below the SC structure lead to a quasi-2D acoustic system, with acoustic dispersions very close to the 2D limit for the low-lying bands (see Supplementary Note 5 for supporting data). The acoustic bands (Fig. 1b) are obtained by solving the acoustic wave equation, $\nabla^2 P = \frac{\rho}{\kappa} \frac{\partial^2}{\partial t^2} P$ ($\rho$ and $\kappa$ are the mass density and bulk modulus, respectively) for the acoustic pressure $P$, using a commercial finite-element software (see Methods). With excellent controllability and versatile measurement methods, macroscopic SCs have manifested themselves as an appealing platform for the study of topological phenomena in classical waves[24–35]. Here we exploit such advantages for the discovery of the symmetry-protected hierarchy of topological multipoles.

**Symmetries and their consequences.** At this point, it is important to point out a number of nontrivial consequences of the $p4g$ symmetry. First, the glide symmetries result in band-sticking effects at Brillouin zone (BZ) boundaries[36–39], leading to double degeneracy for all Bloch bands on the MX and MY lines in the BZ (Fig. 1b). This can be elucidated by introducing the anti-unitary operators $\Theta_i \equiv G_i \mathcal{T}$ ($i = x, y$) with $\mathcal{T}$ being the time-reversal operator. As shown in refs. [36–39], it is straightforward to show that $\Theta_x^2 \Psi_{n,\mathbf{k}} = -\Psi_{n,\mathbf{k}}$ for $k_x = \pi/a$ and $\Theta_y^2 \Psi_{n,\mathbf{k}} = -\Psi_{n,\mathbf{k}}$ for $k_y = \pi/a$, for any acoustic Bloch wavefunction $\Psi_{n,\mathbf{k}}$ (here $n$ and $\mathbf{k}$ are the band index and wavevector, respectively). As an analog to the Kramers theorem, these properties lead to double degeneracy at the BZ boundaries (i.e., the MX and MY lines). In this way, the glide symmetries "glue" the first and second (the third and fourth) bands together. Second, in a $p4g$ crystal, the dipole moment is quantized as $\mathbf{p} = \left(\frac{1}{2}, \frac{1}{2}\right)$, if there are an odd number of bands with parity inversion between the $\Gamma$ and X points, whereas $\mathbf{p} = (0, 0)$, if there are an even number of such bands[9]. From Fig. 1b, we conclude that the first acoustic band gap has a quantized dipole moment of $\mathbf{p} = \left(\frac{1}{2}, \frac{1}{2}\right)$, whereas the second

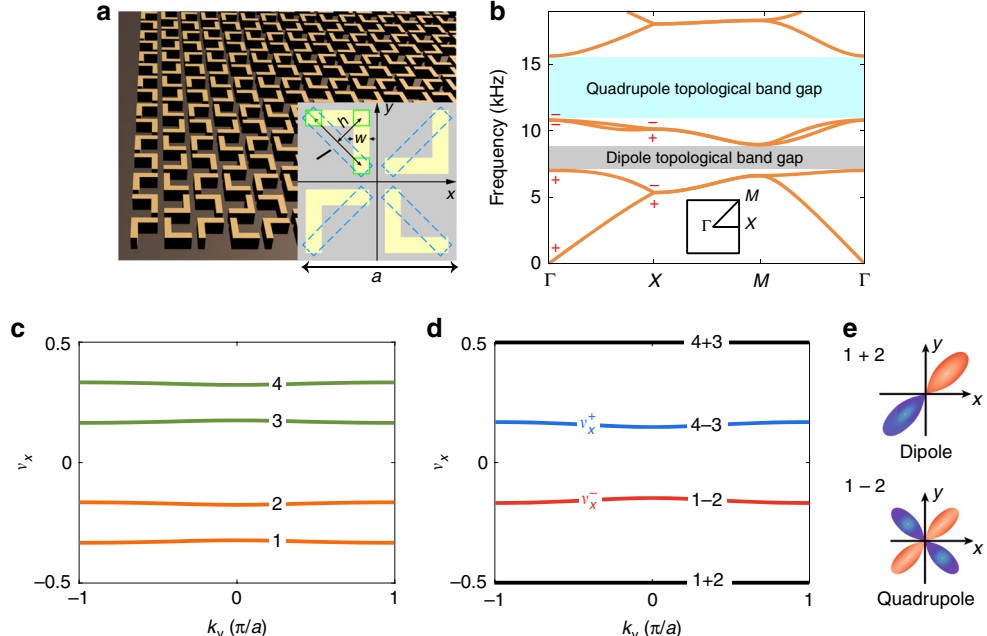

**Fig. 1 Symmetry-protected hierarchy of anomalous topological multipoles in nonsymmorphic sonic crystals. a** A bird's-eye view of the 2D nonsymmorphic sonic crystal cladded by plastic boards from below and above (not shown). The inset illustrates the unit-cell structure with four arch-shaped scatterers made of photosensitive resin (an acoustically rigid material). **b** Acoustic bands and the hierarchy of multipole topological band gaps. The dipole and (anomalous) quadrupole are quantized and protected by the glide symmetries in the $p4g$ group. Inset: Brillouin zone. Symbols $+/-$ represent even/odd parity, respectively, at the $\Gamma$ and $X$ points in the Brillouin zone. **c** Gapped and nondegenerate Wannier bands for the anomalous quadrupole topological gap. **d** Wannier bands for the sum ("$1+2$" and "$4+3$") and difference ("$1-2$" and "$4-3$") sectors. **e** Schematic illustration of the linear combinations of the Wannier orbits in different sectors ("$1+2$" and "$1-2$"), which respectively yield the dipole topology and the anomalous quadrupole topology. Unit-cell geometry parameters are $h = 0.21a$, $l = 0.42a$, $w = 0.1a$, and $a = 2$ cm.

acoustic band gap has a cancelled, vanishing dipole moment. Such a vanishing dipole moment is necessary for the emergence of the QTI. We prove in the Supplementary Notes 1 and 2 that the quadrupole topological index is quantized by the glide symmetries, revealing the picture of symmetry-protected band topology. The resultant anomalous QTI requires at least four bands below the topological band gap and thus a Wannier representation larger than that of the conventional QTIs, which have only two bands below the quadrupole topological band gap. These characteristics are distinctive for the anomalous QTIs protected by the glide symmetries[39].

**Wannier bands for the second band gap**. We use the Wannier bands[8,9] to characterize the quadrupole topology. The Wannier bands are, e.g., the $k_y$ dependence of the Wannier centers $\nu_x^{(n)}(n = 1, 2, 3, 4)$, which are obtained through the Berry phases $\phi_n$ of the first four acoustic bands associated with the Wilson-loop in the BZ ($k_x$ runs from 0 to $2\pi$, for each $k_y$), following the relation $\nu_x^{(n)} = \phi_n/2\pi$ (see Supplementary Note 3 for calculation details). Therefore, the Wannier bands are modulo 1 quantities, as the Berry phases $\phi_n$ are modulo $2\pi$ quantities. In our nonsymmorphic SCs, the Wannier bands are gapped and nondegenerate with two Wannier bands below (above) the polarization gap at $\nu_x = 0$, labeled as "1" and "2" ("3" and "4") in Fig. 1c. The two Wannier bands below the polarization gap can form two different sectors: the "sum" sector that corresponds to the sum of the two Wannier bands (labeled as "$1+2$" in Figs. 1d and 1e) and the "difference" sector for the difference between them (labeled as "$1-2$" in Fig. 1d, e). Similarly, the sum and difference sectors can be defined for the two Wannier bands above the polarization gap (labeled as "$4+3$" and "$4-3$" in

Fig. 1d, e). Interestingly, the difference sector has gapped Wannier bands (see "$1-2$" and "$4-3$" in Fig. 1d, e) and quantized nested Wannier bands, thus yielding a nontrivial quadrupole topological index $q_{xy} = \frac{1}{2}$ (see Supplementary Notes 1 and 3 for details and the rigorous proof). In contrast, the sum sector has gapless Wannier bands and yields quantized, cancelling dipole moments (note that the dipole moments in $1+2$ and $3+4$ sectors cancel with each other). These features signify a novel anomalous QTI protected by the $p4g$ crystalline group. The intriguing nature of this anomalous QTI (Wannier bands and nested Wannier bands, as well as their evolution and transitions, bulk–edge–corner correspondence and other properties) is elaborated detailedly in Supplementary Notes 1–10.

**Interplay between symmetry and topology**. To illustrate the relation between symmetry and topology in our acoustic metacrystals, we study the topological phase transition for the second acoustic gap, which is solely triggered by tuning the geometry of the metacrystal. As shown in Fig. 2a, the continuous geometry transition is as follows: first decrease the arch height $h$ of the arch-shaped scatterers to 0 (indicated by the first three structures), then reduce the arm length $l$ of the scatterers (indicated by the fourth structure), and finally increase the arm width $w$ of the scatterers (indicated by the fifth structure). During this transformation, at $h = 0$, the symmetry of the SC undergoes a transition from the nonsymmorphic $p4g$ group symmetry to symmorphic $C_{4v}$ point group symmetry. We find that the topological transition for the second band gap takes place exactly at such a geometry transition (indicated by the third structure), which elegantly reveals the fundamental interplay between symmetry and topology.

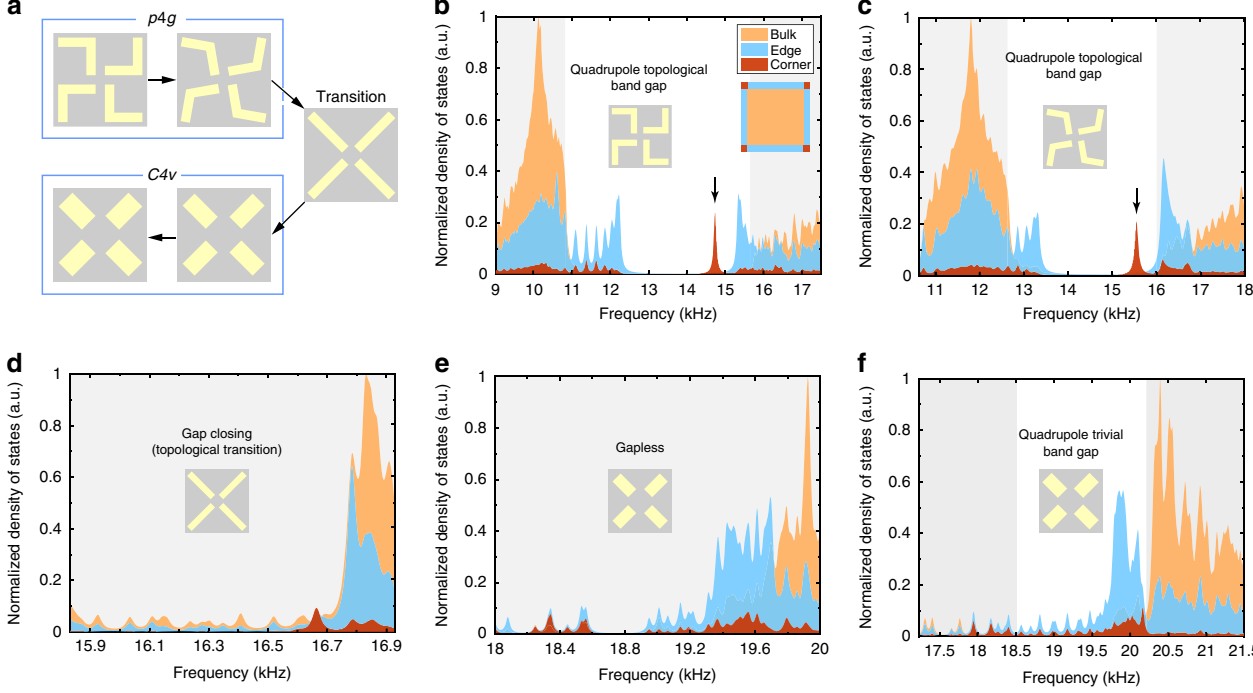

**Fig. 2 Topological phase transitions in the second band gap. a** Topological phase transitions induced by continuously tuning the geometry (indicated by the arrows): first decrease the arch height $h$ to 0 (the first three structures), then reduce the length $l$ (the fourth structure), and finally increase the width $w$ (the fifth structure). The corresponding geometric parameters are $h = 0.21a$, $l = 0.42a$, $w = 0.1a$ for the first structure; $h = 0.15a$, $l = 0.42a$, $w = 0.1a$ for the second structure; $h = 0$, $l = 0.42a$, $w = 0.1a$ for the third structure; $h = 0$, $l = 0.25a$, $w = 0.18a$ for the fourth structure; and $h = 0$, $l = 0.25a$, $w = 0.21a$ for the fifth structure. The first two structures have $p4g$ symmetry, whereas the others have $C_{4v}$ symmetry. **b–f** Calculated local density-of-states for the corner, edge, and bulk regions. The densities-of-states are normalized so that the peak values in the figures are 1. Here, a.u. stands for arbitrary units. The corner, edge, and bulk regions are sketched in the inset of **b**. The calculations are performed on a box-shaped finite structure consisting of $10 \times 10$ sonic crystal unit cells enclosed by hard-wall boundaries. An air channel of width $0.25a$ separates the sonic crystal and the hard-wall boundaries to ensure the physical edges and corners for the acoustic waves.

The local density-of-states for the corner, edge, and bulk regions (illustrated in the inset of Fig. 2b) can be used to reveal the spectral features of the topological transition accompanying the geometry transformation. The calculated densities-of-states (see Supplementary Note 6 for calculation details) are presented in Fig. 2b–f, corresponding to the five structures illustrated in Fig. 2a. It is seen from Fig. 2b, c that the $p4g$ SCs exhibit distinguishable features of the quadrupole topological band gap: the emergence of gapped edge states in the bulk band gap and the sharp peak of corner states within the edge gap. At the transition from the $p4g$ nonsymmorphic symmetry to the $C_{4v}$ point group symmetry, the bulk band gap closes and the corner states merge into the bulk spectrum (see Fig. 2d). For the gapless case (Fig. 2e), there is no spectral feature of the corner states. In the structure with the trivial band gap (Fig. 2f), the edge density-of-states does not exhibit a spectral gap and no corner state is found. Moreover, outside the bulk band gap, the ratio between the corner, edge, and bulk densities-of-states is around 0.1:0.6:1, which is the ratio between the areas of the corner, edge, and bulk regions. Within the bulk band gap, the ratio between the corner and edge states is about the ratio between the areas of the corner and edge regions. These observations indicate that the corner and edge densities-of-states outside the bulk band gap are due to the extended bulk states, whereas the corner density-of-states within the bulk band gap is associated with edge states spreading into the corner regions. The above spectral features verify that the second acoustic band gaps of the $C_{4v}$ SCs do not support corner states and thus indicate their distinct band topology compared with the $p4g$ SCs. Furthermore, as shown in the Supplementary Note 6, the topological transition is also characterized by the close of the

Wannier gap as well as the band structure signatures (including both calculation and comparable experimental results). These results confirm the topological transition from the quadrupole topological band gap to the trivial band gap.

**Quadrupole topological edge and corner states.** We now study the anomalous quadrupole topology by directly measuring the resultant edge and corner states in the second band gap. We first measure the edge states induced by the quadrupole topology. Unlike electronic states in tight-binding models, here the acoustic waves propagate in the air regions among the plastic scatterers in our SCs. To ensure a physical edge, we introduce an air channel of width $0.25a$ between the SC and the hard-wall boundary made of photosensitive resin. This method is commonly used in the study of topological edge states in SCs[31,32,35,40]. The width of such an air channel is so narrow that the waveguide effect of the air channel (which can appear only for frequencies larger than 34 kHz) is excluded for the emergence of the edge states. In fact, the edge and corner states emerge for other widths of the air channel as well. For instance, for the SC in Fig. 1a the edge and corner states emerge for the air channel of width ranging from $0.2a$ to $0.5a$. This demonstrates that the emergence of the corner states is insensitive to the width of the air channel. However, the width of the air channel does affect the frequency of the corner states. We determine the width of the air channel to ensure that the corner states merge into the bulk bands precisely at the topological transition point. In this way, the geometry effect of the air channel on the topological phenomena is minimized.

The experimental dispersions of the edge states are derived from the Fourier transformations of the measured frequency-dependent

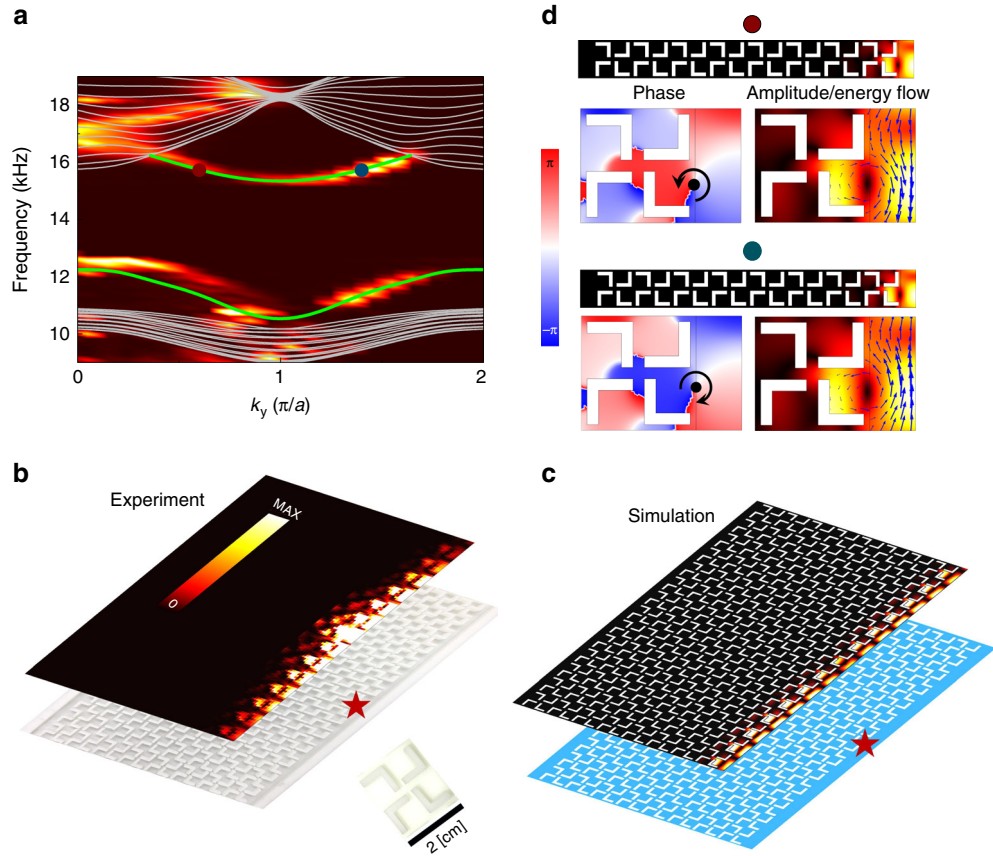

**Fig. 3 Gapped topological edge states induced by anomalous quadrupole in the second band gap. a** Measured (hot color) and calculated (green) dispersions of the acoustic edge states. Gray curves represent calculated acoustic bulk bands. **b, c** Top: measured and simulated acoustic pressure profiles of the edge states launched by a point-like source (indicated by the red stars), respectively. Below: photo of the fabricated sample and the illustration of the structure, respectively. Inset in **b**: photo of a unit-cell. **d** Acoustic pressure profiles for the two edge states marked by the red and blue dots in **a**, with phase, amplitude and Poynting vectors (blue arrows) shown near the edge. Phase vortex centers and phase winding directions are marked by the black dots and arrows, respectively. The edge is formed between the sonic crystal and the hard-wall boundary made of photosensitive resin. Unit-cell geometry parameters are the same as that in Fig. 1.

acoustic pressure profiles along the edge (see Methods). The obtained edge dispersions agree fairly well with the calculation, demonstrating the emergence of gapped edge states due to quadrupole topology (Fig. 3a). In these measurements, the acoustic pressure profiles for the edge states are detected under the acoustic excitation from a point-like source located at the middle of the edge (Fig. 3b). The simulated acoustic pressure profile with a point source excitation under the same conditions is presented in Fig. 3c for comparison. Although the acoustic energy loss is ignored in the simulation, the measured dispersion of the edge states still agrees well with the simulation results, demonstrating that the effect of acoustic energy dissipation is rather small in our experiments. The robustness of the edge states against disorder is studied via numerical simulations in the Supplementary Note 7. Interestingly, we find that the edge states carry finite orbital angular momenta (OAM), which are manifested in two complementary ways: the phase and energy-flow distributions of the acoustic pressure fields. The phase distributions exhibit phase singularities and phase vortices, indicating finite OAM. The two edge states (labeled by the red and blue dots in Fig. 3a), which are time-reversal partners, have opposite phase winding properties, indicating opposite OAM. In addition, the distributions of the energy flow of the acoustic pressure fields (see Methods) also indicate the finite, opposite OAM for those edge states. Figure 3a also shows that the quadrupole topological gap ranges from 10.9 kHz to 15.7 kHz,

reaching a gap-to-mid-gap ratio of 37%. The edge band gap ranges from 12.5 kHz to 15.3 kHz, with a band gap ratio of 20%. These giant topological gaps lead to very strong wave confinement and enhanced wave intensity for the edge and corner states.

We then measure the corner states in a box-shaped, finite-sized structure where the SC is surrounded by hard-wall boundaries (see the inset of Fig. 4a). The calculated acoustic spectrum near the edge band gap is shown in Fig. 4a. Four degenerate acoustic modes, with each of them localized at one of the four corners, emerge in the edge band gap (Fig. 4b, c); this is an important feature of the quadrupole topology[8,9]. The robustness of the corner states against disorders is studied in detail in the Supplementary Note 7.

To confirm the coexistence of the bulk, edge, and corner states in our acoustic system, we measure the frequency-resolved acoustic responses for three different types of pump–probe configurations. We denote these pump–probe configurations as the bulk probe, edge probe, and corner probe, separately. They are illustrated in details in Fig. 4 (see the inset of Fig. 4b). The measured transmission spectra for those pump–probe configurations are shown in Fig. 4b, where we normalize the data to set the peaks of the three curves to unity. The peak of the corner probe lies within the spectral gap of the edge probe, while the peaks of the edge probe lie in the spectral gap of the bulk probe. These spectral features, which are consistent with the calculated acoustic spectrum in Fig. 4a,

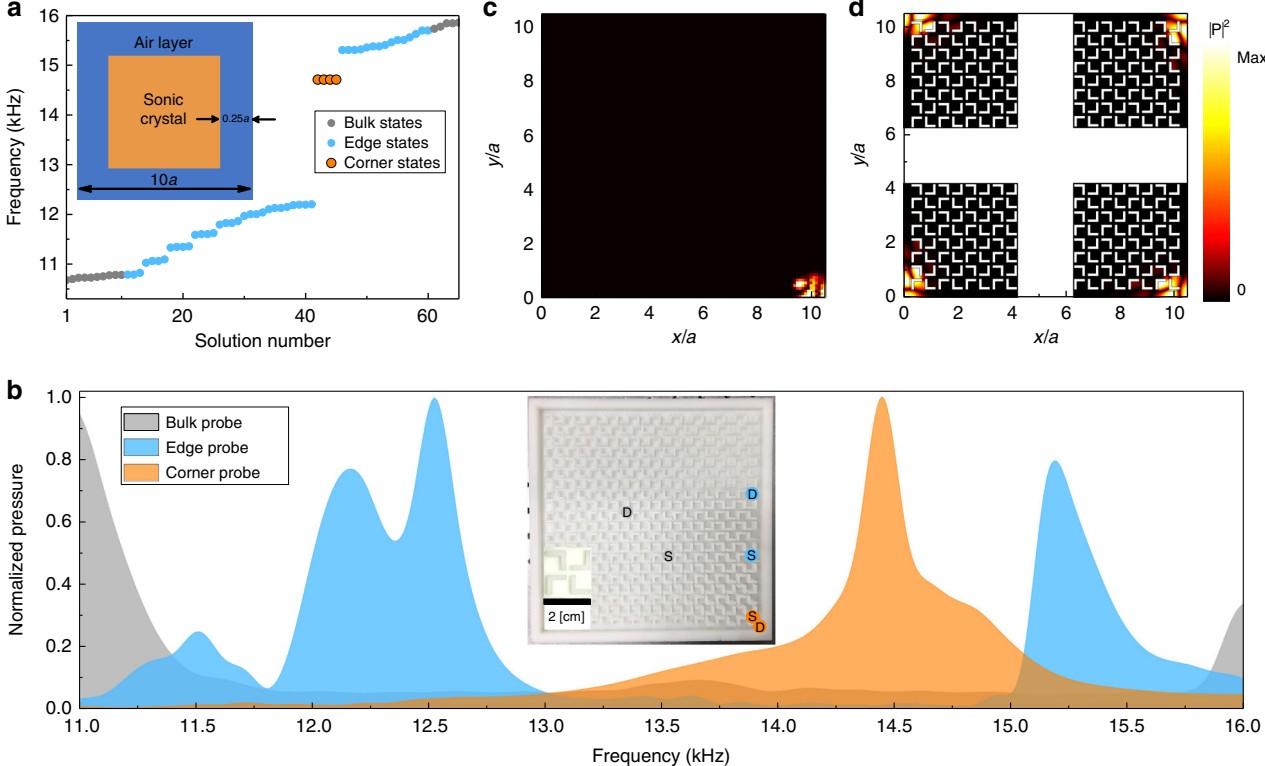

**Fig. 4 Topological corner and edge states in the second band gap due to the anomalous quadrupole. a** Calculated acoustic spectrum for a box-shaped finite structure consisting of 10 × 10 sonic crystal unit cells enclosed by hard-wall boundaries with an air channel of width 0.25$a$ separating them (see the inset for the structure illustration). **b** Frequency-resolved transmission spectra for three pump–probe configurations: bulk probe, edge probe and corner probe. Inset: photo of the sample with a zoom-in photo of the unit-cell. Source ("S") and detector ("D") for each pump–probe configuration (including the bulk, edge, and corner probes) are marked in the inset using the same color as that of the corresponding transmission curve. **c, d** Measured and simulated acoustic pressure profiles of the corner states, respectively. Four degenerate corner states (related by the $C_4$ rotation) are shown in **d**, whereas only one of them is measured in **c**. Unit-cell geometry parameters are the same as that in Fig. 1.

agree well with the simulation of the pump–probe transmission in the Supplementary Note 8, even though we do not include the dissipation in the simulation. Such consistency demonstrates again that the acoustic dissipation is very small in the second acoustic band gap. All those results unambiguously reveal the coexistence of the bulk, edge and corner states as dictated by the multi-dimensional bulk–edge–corner correspondence. In addition, the peak responses of the edge and corner probes are much stronger than the peak bulk-response, reflecting the enhancement of the acoustic field intensity at the edges and corners due to the emergence of the edge and corner states. Bearing in mind that the second band gap has vanishing dipole polarization, these experimental observations are regarded as the key features of the nontrivial quadrupole band topology[8,9] in our SC. Furthermore, we also measure the acoustic pressure profile for one corner at the peak frequency of the corner probe. The measured acoustic pressure profile (Fig. 4c) agrees well with the theoretical acoustic pressure profile obtained from the eigen-mode calculation (Fig. 4d). It is noteworthy that only one corner is measured, as the four corners are connected by the $C_4$ rotation symmetry and are thus equivalent (see Fig. 4d). Figure 4 shows only the absolute value of the acoustic wavefunction, while the real-parts of the corner wavefunctions from both experiments and simulation are shown in the Supplementary Note 9. Together with the pump–probe spectra, those measurements confirm that there is only one localized mode at each corner within the edge band gap, which is another important feature of the QTI[8,9]. It is noteworthy that the measured transmission spectra in Fig. 4b

exhibit a small frequency blue-shift, compared with the calculated acoustic spectrum in the 2D limit in Fig. 4a, which can be associated with the small fabrication imprecision (about ±0.1 mm) and the quasi-2D nature of the experimental system.

**Quantum spin Hall effect in the first band gap**. We now show that the lowest acoustic band gap (i.e., the dipole topological gap) can mimic the quantum spin Hall effect (QSHE) and realize acoustic helical edge states. The double degeneracy at the BZ boundary induced by the glide symmetries provides an instrumental for mimicking pseudospin degeneracy in acoustic systems which have no internal spin (polarization) degree of freedom. As shown later, the acoustic pseudospins are emulated by the OAM of acoustic waves. Following the band-inversion picture in the Bernevig–Hughes-Zhang model for QSHE[41], the nontrivial topology of the quantum spin Hall insulator is characterized by the parity inversion at the high-symmetry points in the BZ. In our SCs, the band inversion can be controlled by rotating the scatterers around the center of each quarter of the unit-cell (see the inset in Fig. 5a for the definition of the rotation angle $\theta$). Differing from Fig. 2, here, the rotation of the scatterers leads to a symmetry transition from the $p4g$ group to the $pgg$ group.

The parity eigenvalues of the lowest two acoustic bands at the Γ, X, Y, and M points are shown in Fig. 5a, b. The parities at the high-symmetry points in the BZ are used to determine the topological properties of the lowest band gap[6,7]. We notice that the parity eigenvalues at the Γ point do not change with the rotation angle. Besides, the X (Y) point always has double

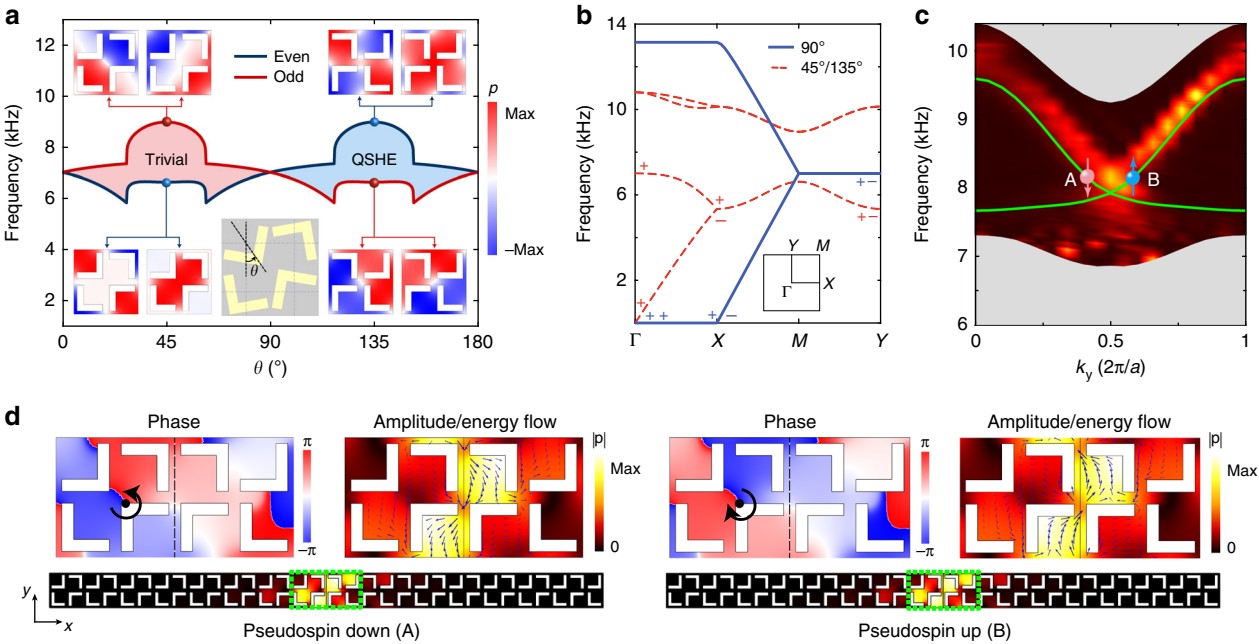

**Fig. 5 Acoustic quantum spin Hall effect and helical edge states in the lowest band gap. a** Evolution of the bulk bands and parities at the M point (including two doublets of even and odd parities) by tuning the rotation angle of the scatterers, $\theta$ (defined in the inset). Acoustic pressure profiles of the doublets at the four marked points are shown as the insets. The parity inversion indicates that the blue region mimics the quantum spin Hall insulator in acoustic systems, while the red region corresponds to the trivial band gap. **b** Acoustic bands for $\theta = 90°$, 45°, and 135°. The symbols $+/-$ represent, respectively, the even/odd parity for the $\Gamma$, X, and Y points for the lowest two bands. **c** Measured (hot color) and calculated (green) dispersions of helical edge states in the lowest band gap at the boundary between two sonic crystals with different $\theta$'s, 45°, and 135°. The gray regions denote the bulk bands. **d** Acoustic pressure profiles for two edge states marked by the red and blue dots in **c**, with amplitude (hot color), Poynting vectors (blue arrows) and phase profiles shown near the boundary (the green-box regions at the bottom panel). Geometry parameters (except for the rotation angle $\theta$) are the same as that in Fig. 1.

degeneracy between two bands with opposite parities. The band inversion with parity switch can take place only at the M point, as shown in Fig. 5a. Such band inversion happens when the rotation angles is an integer of 90°, where the odd- and even-parity bands become degenerate (Fig. 5a, b). It is noteworthy that the flat dispersion along the $\Gamma$X and MY directions at the rotation angle $\theta = 90°$ are due to the fact that the scatterers overlap with each other and form connected "hard walls" along the $y$ direction and forbid wave propagation along the $x$ direction, making the dispersion along $k_x$ flat. This feature, which disappears for smaller scatterers, is not essential for the QSHE as shown in the Supplementary Note 11.

Figure 5a gives the topological phase diagram for the first acoustic band gap. For the rotation angle between 90° and 180°, the band inversion at the M point gives rise to nontrivial band topology (i.e., the acoustic QSHE). In contrast, for the rotation angles between 0° and 90°, the parities at the $\Gamma$ and M points are the same, indicating trivial band topology. The former is regarded as the topological phase because the $\Gamma$ and M points have opposite parities[6,7] for $90° < \theta < 180°$. As the system always has the $C_2$ symmetry, the topological phase diagram has a periodicity of 180°. We further employ a Hamiltonian analysis of the acoustic bands near the M point using the $\mathbf{k} \cdot \mathbf{p}$ method, which reveals that the above two phases are similar to the QSHE and trivial phases in the Bernevig–Hughes–Zhang model[42], respectively (see Supplementary Note 12). An alternative topological theory for the first band gap based on the concept of topological crystalline insulators is presented in the Supplementary Note 13.

The helical edge states emerge in the lowest acoustic band gap at the boundary between the SCs with distinct topology (i.e., QSHE and normal band gap), as shown in Fig. 5c. The calculated and measured (using the same method as in Fig. 3) dispersions of

the acoustic edge states are consistent with each other. It is noticed that the dispersions of the edge states are not well-captured for the low-frequency part. The reason is that the decreased group velocity of the edge states in the low-frequency section leads to longer propagation time and stronger propagation loss and thus yields reduced fidelity of the recorded real-space acoustic pressure profiles and the dispersions obtained from the Fourier transformation of these profiles. The gapless nature of the acoustic helical edge states is guaranteed by the glide symmetry on the edge, which protects the double degeneracy at the $k_y = \pi/a$ point (see Supplementary Note 14 for details)[21]. The robustness of the edge states is elaborated via numerical simulations in the Supplementary Note 15. The pseudospin-momentum-locking feature of the edge states is illustrated in Fig. 5d, where both the phase vortices and the rotating energy flow in the acoustic pressure profiles indicate the finite OAM of the edge states. The acoustic OAM emulates the pseudospins in the acoustic helical edge states and the time-reversed states have opposite pseudospins. In addition, simulation also confirms that the edge states support one-way transport when excited by acoustic sources with finite OAM (i.e., pseudospin-selective excitations; see Supplementary Note 16).

## Discussion
Beside its significance in fundamental science, the discovery of symmetry-protected hierarchy of topological multipoles in nonsymmorphic metacrystals also sheds light on material science. As our symmetry-based approach can realize multipole topological band gaps without engineering to a target tight-binding model, it greatly reduces the difficulties and opens more possibilities in realizing multipole topological phenomena and in exploiting such

phenomena for the practical applications, such as integrated topological waveguides (edge states) and cavities (corner states). Multiplexing topological phenomena in the two topological band gaps can considerably increase the capacity of topologically-protected information processing in a single chip. With the additional advantage that the topological band gaps can be very large, our symmetry-based approach provides an appealing pathway toward multipole topological insulators, which can be generalized to other physical systems (e.g., photonic systems). In addition, our study establishes a bridge between subwavelength metamaterials (artificial functional materials that are useful for a broad range of applications) and topological multipole moments without relying on tight-binding pictures, which are often absent in such metamaterials. When generalized to 3D systems, our symmetry-based approach may offer a possible route towards octupole topological insulators, a novel topological state of matter yet to be discovered.

## Methods

**Experiments**. Our SCs consist of arch-shaped scatterers made of photosensitive resin (modulus 2765 MPa, density 1.3 g cm$^{-3}$). A stereo lithography apparatus (with a fabrication tolerance of roughly 0.1 mm) is utilized to fabricate the samples. The vertical height of the sample is 1 cm. Two boards made of photosensitive resin are used for cladding from the top and the bottom of the sample to form quasi-2D acoustic systems for the frequency range of interest (i.e., <20 kHz). The measured edge-state dispersions are obtained by the following procedure. We first scan the acoustic pressure field distribution along the edge for mono-frequency excitations. An acoustic transducer is placed under the sample to generate acoustic waves, which are further guided into the sample through an open channel (with a diameter of 4 mm) at the bottom of the waveguide. The channel is located at the center of the edge (marked by the red star in Fig. 3b). An acoustic detector (B&K-4939 1/4 inch microphone), whose position can be controlled by an automatic stage, is used to probe the spatial dependence of the acoustic pressure from a circular open window (with a diameter slightly larger than the detector) on the top of the cladding layer. The data are collected and analyzed by a DAQ card (NI PCI-6251). The measured acoustic pressure profiles at different frequencies are then Fourier-transformed to obtain the edge-state dispersions. The Fourier transformation is implemented by using the Matlab built-in function *fft*. The transmission measurements are performed using a similar set-up, but with fixed positions of the source and the detector when the frequency is varied.

In the experimental measurements, the upper board of the waveguide (which is attached to an automatic stage) is required to be able to move freely, but without affecting the stabled samples, to record the acoustic pressure filed data. To accomplish this goal, we leave a tiny air gap (about 1 mm) between the upper board and the samples below it. This treatment might affect our measurements and could be another reason (additional to the fabrication imperfection) that the measurements are slightly deviated from the simulations on frequencies. In addition, the condition for the environment atmosphere that varies upon weather change might also affect the sound speed and the air mass density and is the third reason to the frequency shift between the experiments and the simulations.

The transmission spectra presented in Fig. 4b are normalized by the maximum of each measurement (i.e., the bulk probe, edge probe, and corner probe, respectively), so that they can be plotted at the same quantitative scale. The original, unnormalized transmission spectra are presented separately in the Supplementary Note 8. We find that the corner probe yields a much stronger signal, more than 80 times stronger than the bulk probe, indicating very strong enhancement of the acoustic wave intensity due to the strongly localized subwavelength corner mode.

**Simulation**. Numerical simulations are performed using a commercial finite-element simulation software (COMSOL MULTIPHYSICS) via the acoustic module. The resin objects are treated as hard boundaries. In the eigen-value calculations, the Floquet periodic boundaries are implemented. The projected band structures of the ribbon-like supercells and the band spectrum of the box-shaped supercell are calculated by setting the truncation boundaries as hard boundaries. For the simulated acoustic-pressure distributions of the edge and corner states, the frequency-domain study is performed. A point source, located at the center of the edge (near the corner), is utilized to excite the edge (corner) states. The energy flow is calculated through the time-averaged Poynting vector of the acoustic fields, following $S = -(4\pi\rho f)^{-1}|P|^2\nabla\varphi$, where $\rho$ is the density of air, $f$ is the eigenfrequency, and $|P|$ and $\varphi$ are the amplitude and the phase of the acoustic pressure profile, respectively.

## Code availability

We use the commercial software COMSOL MULTIPHYSICS to perform the simulation and calculations. Request to the details can be addressed to the corresponding authors.

## Data availability

The data that support the findings of this study are available from the corresponding authors upon reasonable request.

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

## Acknowledgements

X.J.Z., Y.T., M.H.L. and Y.F.C. are supported by the National Key R&D Program of China (2017YFA0303702 and 2018YFA0306200), the National Natural Science Foundation of China (Grant Numbers 51902151, 11625418, and 51732006), and the Natural Science Foundation of Jiangsu Province (Grant Number BK20190284). Z.K.L., H.X.W. and J.H.J. are supported by the Jiangsu Province Specially Appointed Professor Funding and the National Natural Science Foundation of China (Grant Numbers 11675116 and 11904060). J.H.J. thanks Arun Paramekanti, Hae-Young Kee, and Gil Young Cho for helpful discussions. He also thanks Sajeev John and the University of Toronto for hospitality where this work is finalized.

## Author contributions

J.H.J. conceived the idea and initiated the project. J.H.J., Y.F.C. and M.H.L. guided the research. J.H.J., Z.K.L. and H.X.W. established the theory. Z.K.L., X.J.Z., Z.X. and H.X.W. performed the numerical calculations and simulations. X.J.Z. and Y.T. achieved the experimental set-up and measurements. X.J.Z., J.H.J. and M.H.L. performed data analysis. All the authors contributed to the discussions of the results and the manuscript preparation. J.H.J., X.J.Z., Z.K.L. and M.H.L. wrote the manuscript and the Supplementary Information.

## Competing interests

The authors declare no competing interests.
