## [Peer Review File · Nature Communications]

Reviewers' comments:

Reviewer #1 (Remarks to the Author):

I read in depth the authors' response letter and the revised version of the article. While the authors have addressed some of my previous comments in the revised article, they have not fully taken into account all of them. More specifically,

1- The authors have added a section in the SM of the revised manuscript about the robustness of the topological edge modes. There, they have claimed that the topological states of their structure can be robust to "various" types of defects. This statement is questionable. It is known that topological states are only protected as long as the symmetry responsible for the topological order of the system is preserved. As a consequence, the system under study is robust to only "certain" kinds of defects or disorder. To test that the robustness of the system is indeed due to the symmetry protected topological phase, the system should be tested against not disorder that (i) breaks the protecting symmetries and (ii) does not break the protecting symmetries.

2- Currently, the robustness of the topological states has only been demonstrated in simulation. However, the advantage and relevance of topological systems versus trivial ones is highlighted better in experiment, where inevitable disorder caused by fabrication could be highly detrimental. I would recommend the authors to verify the robustness of the topological states in experiment as well.

3- The authors have not still experimentally verified that the obtained corner mode has a quadruple moment. This is because they report the absolute value of the field, not its real part. In order to make sure that the obtained higher order topological phase has a quadruple moment, the authors must report the real part of the pressure field distribution.

4- I still do not see any numerical result in the manuscript that cross-validates the experimental transmission spectrum reported in Fig. 4b. In fact, the numerical results provided in Fig. 4a of the revised manuscript reports the eigen-frequency spectrum of the structure, not its transmission spectrum. It is therefore not possible to directly compare the results of these two figures with each other.

Based on these, I cannot recommend the publication of the work, at least in its present form. In order to make a final decision regarding the publication, I need to see all of these issues precisely addressed.

Reviewer #3 (Remarks to the Author):

The authors have successfully addressed all the concerns from the previous version of the manuscript. I would recommend the paper for publication in *Nature Communications* after minor revisions.

I am concerned about the characterization of the mirror-symmetric quadrupole in paragraph 3 of

the introduction (and elsewhere) as “fine-tuned”. First, it is not true that “without the π -flux per plaquette, the quadrupole moment always vanishes” – the quadrupole moment can be quantized by rotation symmetry as long as the flux is non-zero. Furthermore, in the presence of time-reversal symmetry (which the structure in this paper does have) the flux is pinned to either 0 or π , such that the π flux is not in any sense “fine-tuned”. As long as time-reversal symmetry is preserved, which it should always be in a static metamaterial structure, the flux cannot be changed from π without discretely jumping to 0. I recommend that the authors make their point, which is that the mirror symmetric quadrupole is difficult to realize in a sonic crystal due to the π flux, without making the argument that it is because of “fine-tuning”.

The authors have claimed: “the glide symmetries result in band-sticking effects at Brillouin zone (BZ) boundaries, leading to double degeneracy”. It is better they prove this claim explicitly by showing that the Hamiltonian of the system is invariant under the transformation unitary matrix S correspond to the glide symmetries.

It is not clear that the diagram in Fig. 2b is for a finite or infinite system. What is the label and the unit of the x-axis?

We thank the reviewers for the valuable comments and suggestions which help us improve the quality of this manuscript considerably. The main text and Supplementary Information are carefully revised according to those comments and suggestions. New experimental results and simulations are added as suggested by the reviewers. The revisions are marked in the blue color in both the main text and the Supplementary Information. In addition to the revisions made according to the reviewers' comments and suggestions, we also revised the manuscript to meet the editorial requirements of Nature Communications.

Reply to the Reviewer #1

Reviewer's Comments: *"I read in depth the authors' response letter and the revised version of the article. While the authors have addressed some of my previous comments in the revised article, they have not fully taken into account all of them. More specifically, 1- The authors have added a section in the SM of the revised manuscript about the robustness of the topological edge modes. There, they have claimed that the topological states of their structure can be robust to "various" types of defects. This statement is questionable. It is known that topological states are only protected as long as the symmetry responsible for the topological order of the system is preserved. As a consequence, the system under study is robust to only "certain" kinds of defects or disorder. To test that the robustness of the system is indeed due to the symmetry protected topological phase, the system should be tested against not disorder that (i) breaks the protecting symmetries and (ii) does not break the protecting symmetries."*

Our reply: We have carefully revised the sections in the Supplementary Information regarding the robustness of the topological edge and corner states. Our simulation results show that, as remarked by the reviewer, the symmetry-preserving defects have negligible effect on the main spectral features of the edge and corner states, although they may modify some detailed features (which is expected). For instance, if the defects introduce strong perturbation to the system, a new defect mode can be induced. However, the frequency of the corner mode remains stable and the spectral gap of the edge states remains intact. In contrast, the symmetry-breaking defects have much stronger effects on both the corner and edge states as reflected by the transmission spectra in the revised Supplementary Information Notes 7 and 15. These results are consistent with the physical picture of the symmetry-protected topological phases.

Revisions: In the Supplementary Information, we have revised the statements and added detailed discussions about the effects of disorder on the corner and edge states in both Note 7 and Note 15. New figures, Figs. S10, S11 and S21, as from numerical simulations, are added to study the effects of symmetry-preserving and symmetry-breaking defects on the edge and corner states.

Reviewer's Comments: *"2- Currently, the robustness of the topological states has only been demonstrated in simulation. However, the advantage and relevance of topological systems versus*

trivial ones is highlighted better in experiment, where inevitable disorder caused by fabrication could be highly detrimental. I would recommend the authors to verify the robustness of the topological states in experiment as well.”

Our reply: In the revised Supplementary Information, we have included the experimental study on the robustness of the corner state against two types of defects: (i) the symmetry-breaking defects (realized by rotating the plastic structure in the corner unit-cell by 10°) and (ii) the symmetry-preserving defects (realized by amplifying the plastic structure in the corner unit-cell by 10%). The results are presented in a new figure, Fig. S12, in the revised Supplementary Information. For both cases, the experimental results show good consistency with the simulations: (1) the corner states remain within the edge band gap for both cases, (2) the frequency of the corner states is very weakly affected by the symmetry-preserving defects (1%), whereas the frequency of the corner states is considerably affected by the symmetry-breaking defects (up to 6%).

Revisions: In the Supplementary Information, we have added a new figure (Fig. S12) for the experimental results. Related discussions are also added to Note 7 to study the influence of both the symmetry-preserving and symmetry-breaking defects on the corner states.

Reviewer’s Comments: *“3- The authors have not still experimentally verified that the obtained corner mode has a quadruple moment. This is because they report the absolute value of the field, not its real part. In order to make sure that the obtained higher order topological phase has a quadruple moment, the authors must report the real part of the pressure field distribution.”*

Our reply: We agree that the real-part of the acoustic pressure profile contains more information than the absolute value. In the revised Supplementary Information, we present the experimental measurements of the real-part of the acoustic pressure profile for one corner state in Fig. S13. We add Note 8 in the Supplementary Information to discuss the real-part of the acoustic pressure profile for the corner state both numerically and experimentally. The experimental data agree fairly well with the simulation results. We have also tried to measure the acoustic pressure profile for the four corners simultaneously. However, we find that due to the dissipations of the acoustic waves, the four corner states cannot be excited simultaneously by a single source in a large sample where the four corners are well-separated and isolated from each other.

To enable a feasible experimental measurement, we consider a small sample consisting of 4×4 unit cells surrounded by hard walls (with an air gap of the thickness $0.25a$, as in all the measurements and simulations). In such a sample, weak perturbations are introduced near the lower-left and upper-right corners to slightly break the C_4 symmetry down to the C_2 symmetry to fix the sign of the quadrupole moment (as in the seminal papers by Benalcazar, Benervig and Hughes). The perturbations are introduced by the two square-blocks made of photosensitive resin with a width of $w_d = 1$ mm and a distance to the corner of $t_d = 15$ mm (see Fig. R1a for more details). The other geometric parameters are the same as that in Fig. 4 of the main text. We first conduct simulations to see how a point-like source placed at the center can excite the acoustic

modes on the four corners. Here, the mass density of air and the sound velocity are taken as $\rho = 1.21 \text{ kg/m}^3$ and $c = 343 \text{ m/s}$, respectively. As shown in Fig. R1b, the acoustic pressure field distributions from the simulation on the four corners indeed exhibit a quadrupole profile, as suggested by the reviewer. However, we also noticed that the appearance of the quadrupole profile relies on fine-tuning of the perturbation which is very sensitive to the geometry of the square-blocks introduced near the lower-left and the upper-right corners. We suspect that such a fine-tuned phenomenon should not be related to the quadrupole topology and may not even be able to be observed in genuine experiments with defects and dissipation.

Indeed, we could not find such an acoustic field profile in our experiments, although we have tried the finest fabrication and measurements with several samples and set-ups. The observed acoustic field profile exhibits a monopole pattern [see Fig. R2]. In fact, due to the unavoidable dissipation and temperature/humidity dependent sound velocities and air mass density, the fine-tuned acoustic field profile in Fig. R1b can not be observed in experiments. As shown in Fig. R3, with tiny changes in the geometry and/or position of the square-blocks, or the sound velocity, the quadrupole acoustic profile becomes a monopole acoustic profile. From these observations, we have to conclude that the real-space configuration of the quadrupole moment (as manifested by the relative signs of the pressure fields on the four corners) cannot be measured in our acoustic set-ups. Moreover, such a fragile feature cannot be of topological nature.

We emphasize that the above challenges are in fact quite reasonable. Due to the strong localization of the corner states, they are well-separated in a bulk sample. The relative phase of the real-part of the four corner wavefunctions cannot be fixed by any physical mechanism as they do not interact with each other. We note that the quadrupole configuration of the charge density in real space in the papers by Benalcazar, Benervig and Hughes is in fact due to the **band filling** of an electronic system where quadrupole topology induced charge accumulation at the corners can be observed when the system is slightly tuned from C_4 symmetry to C_2 symmetry [termed as “**filling anomaly**” in their papers, see Phys. Rev. B **96**, 245115 (2017), Phys. Rev. B **99**, 245151 (2019)]. Such a fermionic band filling effect is absent in our acoustic system as well as in other bosonic analogs that have been realized in experiments so far.

In fact, what can be measured in acoustic systems are the emergent corner and edge states due to the quadrupole band topology. The nature of the quadrupole topology in this work has been identified through the following aspects: (1) The quadrupole topology is confirmed using gapped Wannier bands and quantized nested Wannier bands in the main text and in the Supplementary Information Note 3. (2) We rigorously prove that the dipole and quadrupole moments are quantized and protected by the two glide symmetries. We also determine the finite dipole moment in the first band gap and the vanishing dipole moment in the second band gap from the parity eigenvalues at the high-symmetry points (Note 2) and the topological crystalline indices (Note 13), as detailed in the Supplementary Information. (3) The spectral signature and the density-of-states evolution during the topological transition also confirms the nature of quadrupole topology: in the quadrupole phase there are corner states emerging in the topological band gap, whereas in the trivial phase there is no such corner states. These results are shown in Figure 2 in the main text

and in Figure S7 and related statements in the Supplementary Information Note 6. (4) The Wannier bands signature during the topological phase transition provides more evidence. The Wannier bands become gapless after the topological transition from the quadrupole topological phase with $P4g$ symmetry to the trivial phase with $C4v$ symmetry (see Supplementary Information Note 6 and Ref. [40]). This signature further confirms the quadrupole topology. (5) A direct, gauge-invariant quantum many-body operator approach is adopted in Ref. [40] to directly calculate the topological quadrupole moment in the $p4g$ wallpaper and the $C4v$ sonic crystals. This approach also confirms that the quadrupole moment in the $p4g$ wallpaper sonic crystals are nontrivial and quantized, whereas it is trivial for the $C4v$ sonic crystals. With all these confirmations from both theory and experiments, it is conclusive that the second band gap is a quadrupole topological band gap with unconventional Wannier bands.

Revisions: The related discussions and statements in the main text are revised e correspondihe corresponding statements and discussions in the main text are revised in Page 13. In the Supplementary Information, we have included a new Note (Note 8) and a new figure (Fig. S13) to present the experimentally measured real-part of one corner state (compared with the simulation). We also show in the figures below the simulated real-space configuration of the quadrupole moment in a small sample and their extreme sensitivity on the geometry of the square-blocks and the sound velocity in air.

Fig. R1 | Simulated real-space configuration of the quadrupole corner states as excited by a point source in the center for a small sample with 4×4 unit cells. a, The schematic of the corner structure under consideration. Two small perturbations (indicated by the tiny orange square-blocks; see also the zoom-in) are introduced to reduce the $C4$ symmetry to the $C2$ symmetry to fix the sign of the quadrupole moment. **b,** Simulated acoustic pressure fields under a point source placed at the center of the structure. The excitation frequency is tuned to the corner resonance, 14.72 kHz.

Fig. R2 | Measured real-space configuration of the quadrupole corner states as excited by a point source in the center for a small sample with 4×4 unit cells. The structure is fabricated using high-resolution (0.1 mm resolution) 3D-printing technology. The figure displays the measured acoustic pressure fields under a point source placed at the center of the structure. The excitation frequency, 14.67 kHz, is very close to the resonance frequency of the corner states from the simulation in Fig. R1.

Fig. R3 | Effects of tiny geometry modification of the square-blocks and the slight change of the sound velocity on the acoustic field distributions on the corners. These are simulation results. In the simulation in

a-c only tiny changes of the geometry/position of the square-blocks are introduced. In the simulation in **d** only the sound velocity in air is slightly modified. From all these simulations, one can see that with tiny changes of the physical/geometric parameters, the acoustic pressure field profiles of the corner states change from the quadrupole shape in Fig. R1 to the monopole shape. These simulation results indicate that the quadrupole shape in Fig. R1 is not a reflection of the quadrupole topology, but due to some fine-tuned mechanisms.

Reviewer's Comments: *"4. I still do not see any numerical result in the manuscript that cross-validates the experimental transmission spectrum reported in Fig. 4b. In fact, the numerical results provided in Fig. 4a of the revised manuscript reports the eigen-frequency spectrum of the structure, not its transmission spectrum. It is therefore not possible to directly compare the results of these two figures with each other."*

Our reply: In the revised Supplementary Information Note 9, we include the simulation corresponding to the experimental transmission measurements in Fig. 4b (see Fig. S16 and related paragraphs in Note 9 in the Supplementary Information). The key spectral features in Fig. S16 agree quite well with those in Fig. 4b of the main text. For instance, the emergence of the edge transmission peaks in the bulk band gap and the corner transmission peak within the band gap of the edge states is clearly visible. The frequencies of the corner states, the edge bands, and the bulk bands are comparable with the the experimental transmission spectra as well. Some detailed features are different. For instance, the corner resonance in the simulation is at 14.7 kHz, while in the experiments it is 14.5 kHz. In addition, the edge spectral gap in the simulation is slightly larger than the measured edge spectral gap in experiments. Most notably, the resonances for both the edge and corner states are much broader in the experiments than in the simulation. These differences are mainly because of the following reasons: (1) In the simulation, we use point sources and point detectors for the excitation and detection of acoustic waves. However, in experiments, the sources and the detectors have finite size (their genuine geometric features are hard to model in the simulation). (2) Acoustic waves have dissipation during the propagation processes in realistic ambient environment, which we did not include in the simulation. (3) We do 2D simulations, while the experiments is done in a quasi-2D system. (4) The speed of sound and the mass density of air may be different in the laboratory environment compared with the ideal set-up in the simulation.

Revisions: In the revised Supplementary Information, we provide the simulation results for the transmission spectrum with the same pump-probe set-up as in Fig. 4b of the main text (except a few idealizations). A new figure (Fig. S16) is added to Note 9. Descriptions and discussions on the similarity and difference between the simulation and the experimental measurements are also added in Note 9 and in the main text (see Page 9).

Reply to the Reviewer #3

Reviewer's Comments: *"The authors have successfully addressed all the concerns from the previous version of the manuscript. I would recommend the paper for publication in Nature Communications after minor revisions. I am concerned about the characterization of the mirror-symmetric quadrupole in paragraph 3 of the introduction (and elsewhere) as "fine-tuned". First, it is not true that "without the π -flux per plaquette, the quadrupole moment always vanishes" – the quadrupole moment can be quantized by rotation symmetry as long as the flux is non-zero. Furthermore, in the presence of time-reversal symmetry (which the structure in this paper does have) the flux is pinned to either 0 or π , such that the π flux is not in any sense "fine-tuned". As long as time-reversal symmetry is preserved, which it should always be in a static metamaterial structure, the flux cannot be changed from π without discretely jumping to 0. I recommend that the authors make their point, which is that the mirror symmetric quadrupole is difficult to realize in a sonic crystal due to the π flux, without making the argument that it is because of "fine-tuning"."*

Our reply: In the revised manuscript, we have modified the related statements according to the reviewer's suggestion.

Revisions: The revisions are made throughout the manuscript to avoid the argument of "fine-tuning" for the π flux lattice model and its realizations.

Reviewer's Comments: *"The authors have claimed: "the glide symmetries result in band-sticking effects at Brillouin zone (BZ) boundaries, leading to double degeneracy". It is better they prove this claim explicitly by showing that the Hamiltonian of the system is invariant under the transformation unitary matrix S correspond to the glide symmetries."*

Our reply: In the revised manuscript, we add arguments and an analytical proof for the band-sticking effects at the BZ boundaries due to the glide symmetries. In our sonic crystals, the acoustic band structure is calculated numerically by solving the acoustic wave equation using the finite-element method, where the Hamiltonian does not explicitly exist. Nevertheless, the symmetries of the sonic crystals impose constraints on the degeneracy and the Bloch wavefunctions of the acoustic waves.

Revisions: The revisions are made in the main text page 5.

Reviewer's Comments: *"It is not clear that the diagram in Fig. 2b is for a finite or infinite system. What is the label and the unit of the x-axis?"*

Our reply: Considering that Fig. 2b contains too much information that is not able to be elaborated in the main text. We move it into the Supplementary Information. To make Fig. 2 more transparent and understandable, we revised Fig. 2a to contain 2 unit-cells with P4g symmetry and 2 structures with C4v symmetry as well as a structure sitting on the transition point. The local density of states

for the bulk, edge and corner regions (for a finite structure with 10×10 unit-cells and a physical boundary between the sonic crystal and the hard-wall boundaries) for all 5 structures are presented in Fig. 2b-2f, along the geometry transition illustrated in Fig. 2a, i.e., first reduce the arch-height h to 0, then decrease the length of the block, and finally increase the width of the block. The local density of states reflect directly that such a geometry transition triggers a topological transition in the second band gap from quadrupole topological band gap to trivial band gap: The $p4g$ structures host gapped edge states and in-gap corner states, whereas the $C4v$ structures are either gapless or have a trivial band gap which do not support corner states. The topological transition takes place exactly at the geometry transition point between the $p4g$ and $C4v$ symmetries, which elegantly reflects the fundamental interplay between symmetry and topology.

Revisions: The revisions are made to Fig. 2 and the related discussions in pages 6 and 7 in the main text, and the Supplementary information Note 6 and Figure S7.

Reviewers' comments:

Reviewer #1 (Remarks to the Author):

I read in depth the response letter of the authors of this paper.

They have now studied, in theory and experiment, the robustness of the corner states against both symmetry preserving and symmetry breaking defects. I find this part satisfying.

One major point that I feel is left largely unanswered is related to the fact that "due to the dissipations of the acoustic waves, the four corner states cannot be excited simultaneously". Because of this, they could not see the quadruple moment of the corner state in experiment. This is a bit of a pity since this is an important prediction of the model that is not validated by the experiment. My main worry is that the presence of losses could make the system strongly non-Hermitian, which changes the definition of the topology of the system, and forcing the quadrupole to break down into four independent monopoles.

Regarding the glide symmetry : These symmetries are the ones that protect the higher order corner states. It would be good that the authors rigorously demonstrate that the Hamiltonian of the system respects these symmetries. The authors mention that their acoustic system cannot be modeled with a simple tight binding model, so the Hamiltonian is not simply available. However, I cannot agree, I would expect that a $k.p$ model near the band edges would help removing this caveat.

Reviewer's comments: *One major point that I feel is left largely unanswered is related to the fact that "due to the dissipations of the acoustic waves, the four corner states cannot be excited simultaneously". Because of this, they could not see the quadruple moment of the corner state in experiment. This is a bit of a pity since this is an important prediction of the model that is not validated by the experiment. My main worry is that the presence of losses could make the system strongly non-Hermitian, which changes the definition of the topology of the system, and forcing the quadrupole to break down into four independent monopoles.*

Our reply: We thank the reviewer for raising this point. Here, we would like to emphasize that the direct consequence of the quadrupole topological band gap is the emergence of the gapped edge states and in-gap corner states, as observed in the seminal experimental works [Nature 555, 342–345 (2018); Nature 555, 346–350 (2018); Nature Physics 14, 925–929 (2018)], as well as in our experiments. In these works, which study the classical wave systems, the “quadruple moment of the corner state” was never reported and is unlikely to be observed. Specifically, in the paper [Nature 555, 342–345 (2018)], Fig. 3c shows that the distribution of the corner states is even not C_4 symmetric, which thus cannot yield the quadrupole configuration of the corner states. In another paper [Nature 555, 346–350 (2018)], Fig. 3f shows that the four corner modes in their experiments are NOT degenerate, which means that they cannot emerge as a “quadrupole configuration”. In the paper [Nature Physics 14, 925–929 (2018)], the authors studied only one corner (see Fig. 2b in the paper), as in our experimental study.

In fact, the "quadrupole moment of the corner state" is impossible for bosonic or classical wave systems (as elaborated in details in our last reply), since it is a band filling effect, which can only emerge in electronic systems with the fermionic band filling. The **band filling effect** has been clarified in the seminal work that proposes the quadrupole topological insulator [Science 357, 61–66 (2017)]. Specifically, the quadrupole configuration of the corner charge in the paper [Science 357, 61–66 (2017)] comes from the energy splitting of the corner states (due to the slight breaking of the C_4 symmetry designed in their calculations) and the energy level filling (up to the Fermi energy) instead of the corner wavefunctions. Even in such electronic systems, the corner states are independent from each other, since they can be very far away from one another in a macroscopic system. The electronic wavefunctions CANNOT maintain a coherent quadrupole configuration over macroscopic scales, which is against common sense. Therefore, it is unlikely to observe the quadrupole configuration in the corner wavefunctions, which is not at all related to the quadrupole topology. In the bosonic or classical wave systems

(such as our acoustic system), the wavefunctions (i.e., the field distributions) are the main measurable physical quantities (the other quantity is the transmission), while the quadrupole configuration of the corner charge, exclusively in electronic systems, is not expected to appear.

In our last reply, we used the simulation results to show that in certain situations with fine-tuning, the quadrupole configuration of the corner state does appear. However, changing the sound velocity in air (or other geometric parameters) slightly can kill the quadrupole configuration. This again demonstrates that such a fragile feature cannot be a smoking-gun feature of the quadrupole topology (because topology should be a robust feature). In our experiments, there are many factors that could kill such a non-topological feature, such as the temperature, humidity and small dissipation. However, these factors do not affect the quadrupole topology in our system, which is manifested unambiguously by the gapped edge states and in-gap corner states as well as numerical identification of the Wannier band polarizations.

To avoid the potential mixing of the concepts of “quadrupole topological insulator” (that describes a quadrupole topological band gap and the edge and corner states within the band gap) and the “real-space quadrupole moment”, we removed all the terms of “quadrupole moment” in the revised manuscript. What we predicted in theory and observed in experiments is the acoustic analog of the “quadrupole topological insulator” and the resultant edge and corner states. Our study has nothing to do with the “real-space quadrupole moment”.

In our experimental systems, there do exist dissipation effects. However, the phase coherence length of the acoustic waves is quite long (that’s why we can communicate easily using verbal languages in our daily life). In fact, it is because of the long phase coherence length, we can scan the acoustic wave profiles for the bulk and edge states in the entire sample to extract their dispersions using the Fourier-transformation (see Fig. 3 in the main text; the measured edge states dispersion agrees quite well with the simulation which do not include the acoustic loss). In addition to the gapped edge states in Fig. 3, we observed the in-gap corner states in Fig. 4, which also agree well with the simulation without loss. From these experimental results, it is found that the small dissipation (or the non-Hermitian effect it introduced) of the acoustic waves indeed does not lead to any gap-closing on the edge or in the bulk, nor does it affect the emergence of the gapped edge states and the in-gap corner states due to the quadrupole topology. In the revised manuscript, we have included the statements to clarify that the small dissipation in experiments does not affect the observation of the gapped edge states

and the in-gap corner states due to the quadrupole topology.

Reviewer's comments: *Regarding the glide symmetry: These symmetries are the ones that protect the higher order corner states. It would be good that the authors rigorously demonstrate that the Hamiltonian of the system respects these symmetries. The authors mention that their acoustic system cannot be modeled with a simple tight binding model, so the Hamiltonian is not simply available. However, I cannot agree, I would expect that a k.p model near the band edges would help removing this caveat.*

Our reply: We thank the reviewer for this suggestion. In fact, we already have provided a k.p Hamiltonian in the Supplementary Note 12, which respects the glide symmetries. To make the geometric symmetry clearer, in the revised manuscript we have also added a coordinate system in the structure of the unit-cell in Fig. 1a.

Reviewers' Comments:

Reviewer #1:

Remarks to the Author:

I think the review process can be concluded on these interesting discussions about the quadrupolar nature of the corner states, which are mentioned but don't need to be fully elucidated here (as the author said, this is not inherent to this particular study).

As for the kp model, I agree that it is sufficient to justify the symmetry arguments of the authors. I can therefore safely recommend publication of the work.

Reply to the referee #1

Comments & remarks of the referee: *I think the review process can be concluded on these interesting discussions about the quadrupolar nature of the corner states, which are mentioned but don't need to be fully elucidated here (as the author said, this is not inherent to this particular study). As for the kp model, I agree that it is sufficient to justify the symmetry arguments of the authors. I can therefore safely recommend publication of the work.*

Our Reply: We thank the referee for the recommendation! We are glad that these interesting discussions have been raised which help us and the community to understand better the nature of the classical-wave analogs of quadrupole topological insulators: the similarity and difference between the electronic quadrupole topological insulators and their classical-wave analogs; which effects can be observed and which cannot in a classical-wave quadrupole topological insulator.